# DIVERSITY-AWARE TRAINING FOR TEST-TIME SCALING

## ABSTRACT

Test-time scaling for large language models (LLMs) is a recognized effective approach to improving performance. However, when increasing test-time computation, the performance gains grow progressively smaller. This is largely due to the tendency of independent reasoning attempts to collapse into similar incorrect solutions. Existing approaches to enhancing reasoning diversity mainly focus on token-level diversity, which fail to capture reasoning-level diversity and introduce hallucinations. To this end, we introduce REPRISM, a novel framework designed to act like a Reasoning Prism, guiding models to explore a spectrum of distinct and valid reasoning paths from a single input. First, we construct training data where each prompt is associated with multiple diverse yet correct answers. Additionally, we introduce noise embeddings into special tokens as implicit diversity signals, teaching the model to recognize these embeddings as indicators of diverse reasoning paths. We validate REPRISM on 9 challenging benchmarks across Math, Code, and Agent tasks, where it increases the models' pass@N accuracy by up to 6.4%, 1.1%, and 0.5% on Math, Code, and Agent tasks, respectively. Moreover, we demonstrate that the reasoning diversity instilled by REPRISM provides a superior foundation for Reinforcement Learning (RL). REPRISM not only furnishes a richer exploration space that leads to enhanced performance gain from RL, but also prevents the collapse of reasoning diversity during RL training.

## 1 INTRODUCTION

Test-time scaling (TTS) i.e., harnessing increased compute at inference by sampling more candidate solutions, has recently emerged as an effective approach to improving performance of large language models (LLMs) (Snell et al., 2024; Muennighoff et al., 2025; Hooper et al., 2025). A central metric for TTS effectiveness is sampling efficiency: how the success rate improves as the sample budget increases (Chen et al., 2025a). Sampling efficiency is particularly significant in domains where solutions are verifiable, such as math (Ren et al., 2025), code (Chen et al., 2021), and games (Silver et al., 2017), as the primary goal is to discover a correct solution or optimizes for the best outcome. TTS also boosts the efficiency of rejection-sampling fine-tuning (Yuan et al., 2023).

However, despite the potential of TTS, the marginal performance gains decrease rapidly as more computation is allocated, hurting overall sampling efficiency (Wang et al., 2024b; Shao et al., 2024). This is largely due to the tendency of independent reasoning attempts to collapse into similar solutions, leading to wasted samples on repeated errors (Chen et al., 2025c; Huang et al., 2023). The key to unlocking the potential of TTS, therefore, lies in enhancing the diversity of the generated texts.

Existing approaches to enhance reasoning diversity (Geiping et al., 2025; Zhang et al., 2025; Chen et al., 2025b) mainly focus on prompt engineering (Naik et al., 2023) and token-level decoding strategies (Vijayakumar et al., 2016; Ippolito et al., 2019; Meister et al., 2023). Prompt-based methods rely on human prior experience which can't generalize to different domains. Token-level decoding methods achieve only syntactic variation without ensuring logical diversity in the reasoning process. A core deficiency shared by these approaches is that they are inference-time patches and the models were never trained to produce diverse reasoning, therefore remain agnostic to the concept of diversity and lack a control mechanism for it.

To this end, we introduce REPRISM, Reasoning Prism, which is designed to guide LLMs in exploring a diverse spectrum of distinct reasoning trajectories from a single input, as conceptually illustrated

Figure 1: **(Left)** Comparison of SFT and REPRISM. **(Right)** The average performance of REPRISM, SFT, and base models under different sampling budgets on different tasks.

in Figure 1 (left). Our methodology first involves the construction of specialized training datasets where each prompt is paired with multiple diverse but correct answers. Subsequently, we incorporate noise embeddings into special tokens during both the training and inference phases. This technique effectively trains the model to interpret these noise patterns as implicit indicators for generating different reasoning paths from the same input. We also provide theoretical analyses that prove the stability of our method and underpin the principled nature of our framework in enabling such diverse yet controlled generation.

We validate the efficacy of RePrism through extensive experiments on 9 challenging benchmarks spanning Math, Code, and Agent tasks. Under equivalent compute budgets, REPRISM consistently achieves state-of-the-art sampling efficiency, increasing Pass@N accuracy by up to 6.4% on Math, 1.1% on Code, and 0.5% on Agent benchmarks. Figure 1 (right) depicts the comparison of sampling efficiency of REPRISM and SFT models. Ablation studies corroborate the effectiveness of our proposed diverse data sampling and noise embedding mechanisms.

Furthermore, in Reinforcement Learning (RL) training, a lack of rollout diversity leads to reward scores lacking variance, and causes saturation (Cui et al., 2025; Lin et al., 2025). Beyond improving test-time sampling, we find that the diversity fostered by REPRISM enhances exploration and subsequent performance in RL. The models trained with our method provides a more diverse starting point for RL fine-tuning, allowing a much richer variety of valid reasoning trajectories from the outset. We also find that this enhanced exploration capability can be sustained throughout the RL process with REPRISM, whereas models trained with standard methods suffer a collapse in sampling performance.

Our main contributions are summarized as follows:

- We introduce REPRISM that enhances reasoning diversity through specialized training data and a noise-embedding mechanism, enabling implicit control of distinct reasoning paths.
- We did holistic experiments on Math, Code, and Agent domains, results show that REPRISM significantly improves Pass@N accuracy and sampling efficiency.
- We show that models trained with REPRISM provide a robust foundation for RL. It substantially expands the exploration space and maintains models' sampling performance.

## 2 RELATED WORKS

**Test-Time Scaling of LLMs.** Test-time scaling (Snell et al., 2024; Muennighoff et al., 2025; Hooper et al., 2025) aims to improve models' performances by investing more computational resources during the inference phase, rather than solely relying on computational investment during pre-training. Small models are possible to achieve or even surpass the performance of larger models with fewer computational resources under specific conditions (Anil et al., 2023; Hendrycks et al., 2021; Qu et al., 2024; Lightman et al., 2023; Wang et al., 2023). However, the effectiveness of test-time scaling can be limited when scaling computing encounters diminishing returns, especially when exponentially growing sampling budgets may not necessarily yield linear performance improvements (Huang et al., 2023; Stechly et al., 2023; Valmeekam et al., 2023; Wang et al., 2024b; Shao et al., 2024).

**Diverse Generation of LLMs.** Supervised fine-tuning results in models generating deterministic and repetitive outputs (Kim et al., 2024; O'Mahony et al., 2024; Shypula et al.), and increasing their

diversity has been a long-standing challenge (Li et al., 2015; Vijayakumar et al., 2016). Current methods mainly focus on decoding algorithms (Holtzman et al., 2019; Vijayakumar et al., 2016; Meister et al., 2023), prompt and pipeline engineering (Naik et al., 2023; Wang et al., 2024a; Summers-Stay et al., 2023; Li et al., 2025), and specially designed loss terms that encourage token-level diversity (Jiang et al., 2019; Li et al., 2024). Yu et al. (2024) considered reasoning-level diversity but is limited to tasks that have step-level rewards. Moreover, output diversity is also crucial for RL training (Lin et al., 2024; Hong et al., 2018; Eysenbach et al., 2018), as it drives agents to explore more broadly to discover better policies, and autonomously learn finer reusable skills.

**Adding Noise to LLMs' Embeddings.** Some have improved LLMs' performance by adding noise to the embedding during training (Jain et al., 2023; Zhu et al., 2019). However, their usage of noise is a kind of regularization (Bishop, 1995), while in our work it's a signal of diversity.

## 3 METHODOLOGY

Our methodology aims to enable LLMs to generate diverse yet correct reasoning trajectories. We first construct diverse training datasets using submodular optimization to sample varied reasoning paths (Section 3.1), then train and do inference with the model with noisy special-token embeddings (Section 3.2). We provide analysis on our training target to understand its impact (Section 3.3).

### 3.1 DIVERSE REASONING TRAJECTORY SAMPLING VIA SUBMODULAR OPTIMIZATION

Following prior work on diversity as feature coverage in summarization and data selection (Filatova & Hatzivassiloglou, 2004; Takamura & Okumura, 2009; Lin & Bilmes, 2011), we define the diversity of a subset $A \subseteq S$ as the number of distinct reasoning features it covers:

$$f(A) = \left| \bigcup_{s_a \in A} F(s_a) \right|, \tag{1}$$

where $F(s_a)$ denotes the set of reasoning features (e.g., n-grams in language, AST elements in code) contained in trajectory $s_a$.

To maximize this diversity under a budget constraint, we adopt a two-stage framework. First, for each input, we collect a candidate set of $m$ trajectories, denoted as $S = \{s_1, \ldots, s_m\}$. Second, we employ a submodular subset optimization strategy (Krause & Golovin, 2014) to extract a fixed-size subset $A \subseteq S$ of $n$ trajectories that maximizes $f(A)$.

We use a greedy algorithm to maximize $f(A)$ subject to $|A| = n$. Starting with an empty set $A_0 = \emptyset$, at each step $t$ ($t = 0, \ldots, n-1$), we select the trajectory $s_{t+1}$ that offers the largest marginal gain:

$$s_{t+1} = \underset{s_i \in S \setminus A_t}{\arg\max} \big( f(A_t \cup \{s_i\}) - f(A_t) \big). \tag{2}$$

Let $U_t = \bigcup_{s_a \in A_t} F(s_a)$ denote the features already covered. The criterion simplifies to:

$$s_{t+1} = \underset{s_i \in S \setminus A_t}{\arg\max} \big| F(s_i) \setminus U_t \big|. \tag{3}$$

This algorithm guarantees a $(1 - 1/e)$ approximation to the optimal feature coverage achievable with $n$ samples (Nemhauser et al., 1978). Appendix A further explains it.

To empirically validate the effectiveness of diversity sampling with submodular optimization methods, we authors conducted a human evaluation to verify the diversity of sampled data. Specifically, we randomly sampled 32 queries and their corresponding 4 outputs in our sampled code and math dataset. Human evaluators then annotate how many different solutions they contain ($\geq$ 3, 2, or 1). The results of this evaluation are presented in Table 1, where the numbers are the percentage of queries falling into each of the categories. The table shows diversity sampling with submodular optimization consistently yields a higher percentage of queries with $\geq$ 3 distinct solutions than the baseline w/o submodular optimization where we randomly sample 4 trajectories for each query, confirming its effectiveness. A detailed evaluation protocol is available in Appendix F. A further check shows that queries with low solution diversity were typically simple problems that inherently have few distinct algorithmic solutions.

Table 1: Comparison of solution diversity in training data with different sampling methods, evaluated by humans.

| Datasets | $\geq 3$ | 2 | 1 |
|---|---|---|---|
| Code | 34.4 | 37.5 | 28.1 |
| w/o submodular optimization | 25.0 | 37.5 | 37.5 |
| Math | 42.8 | 39.3 | 17.9 |
| w/o submodular optimization | 28.1 | 37.5 | 34.4 |

### 3.2 NOISY SPECIAL-TOKEN EMBEDDINGS AS IMPLICIT DIVERSITY SIGNALS

While traditional SFT learns a direct mapping between inputs and outputs, it does not account for the need to generate diverse yet correct outputs for the same input. To foster diversity-controllable sampling, we hope to control the models to produce different valid reasoning paths.

To achieve this, we introduce Gaussian noise perturbations to the embeddings of special input tokens during both training and inference. Let $x = (x_1, \ldots, x_L)$ be an input sequence of token IDs. The standard sequence of embedding vectors for $x$ is $\mathbf{E}_x = (\mathbf{e}_{x_1}, \ldots, \mathbf{e}_{x_L})$. We define a corresponding noise sequence $\mathbf{N}_x = (\mathbf{n}_{x_1}, \ldots, \mathbf{n}_{x_L})$. For each position $j$, the noise vector $\mathbf{n}_{x_j}$ is sampled from $\mathcal{N}(\mathbf{0}, \sigma^2 \mathbf{I})$ if $x_j$ is a special token, and $\mathbf{n}_{x_j} = \mathbf{0}$ if $x_j$ is not a special token. This targeted noise injection perturbs the input representation at specific special token positions, thereby stimulating diverse model outputs while keeping the core semantic information of the input sequence $x$ intact.

The effective input embedding sequence to the model, $\mathbf{E}'_x$, is the element-wise sum of the two parts:

$$\mathbf{E}'_x = \mathbf{E}_x + \mathbf{N}_x = (\mathbf{e}_{x_1} + \mathbf{n}_{x_1}, \ldots, \mathbf{e}_{x_L} + \mathbf{n}_{x_L}). \tag{4}$$

The training objective is formulated to learn from these noise-perturbed input representations:

$$\mathcal{J}(\theta) = \mathbb{E}_{(x,\mathbf{N})} \left[ -\log p_\theta(y(x, \mathbf{N}) | \mathbf{E}'_x) \right]. \tag{5}$$

Here, $x \sim \mathcal{D}_x$ represents sampling from the input distribution and $y(x, \mathbf{N}) \sim \mathcal{D}_y^x$ denotes one of the diverse correct answers for question $x$ (i.e., an element from set $A$ as described in Section 3.1).

By training the model to predict different $y \in A$ for the same $x$ but different instances of $\mathbf{E}'_x$, the model implicitly learns to associate different noise patterns with different valid reasoning paths. Consequently, at inference time, applying varied noise sequences allows for guiding the model towards generating diverse responses for the same input $x$.

To ensure practical scalability, we also provide an efficient implementation of this mechanism within pyTorch (Paszke et al., 2019) and vLLM (Kwon et al., 2023) frameworks.

### 3.3 TRAINING TARGET ANALYSIS

We conduct theoretical analyses to answer two questions: (1) Does injecting noise into special-token embeddings destabilize the language model? (2) How does our training formulation facilitate controllable diversity at inference time?

**Embeddings.** For the embeddings, the introduction of Gaussian noise acts as a form of regularization. Formally, let the vocabulary size be $V$ and the embedding dimension be $D$. The embedding matrix is denoted as $\mathbf{E} \in \mathbb{R}^{V \times D}$. $\mathbf{E}'$ is the embedding added with randomly sampled Gaussian Noise. Analyzing the expected loss function $\mathbb{E}_{\mathbf{n}}[L(\mathbf{E}')]$ reveals that it introduces an regularization term:

$$\mathbb{E}_{\mathbf{n}}[L(\mathbf{E}')] = L(\mathbf{E}) + \frac{\sigma^2}{2} \sum_{i \in \mathcal{S}} \text{tr}(\mathbf{H}_i) + O(\sigma^3), \tag{6}$$

where $\mathbf{H}_i$ is the Hessian of the loss function with respect to the embedding $\mathbf{e}_i$. This term, proportional to the trace of the Hessian, is a style of regularization (Bishop, 1995) by penalizing high-curvature regions of the loss landscape. Therefore, adding noise to the embedding wouldn't cause any instability. A detailed derivation of this result is in Appendix B.

Table 2: Pass@N performance on math benchmarks. We mark the column of highest pass that symbolizes test-time scaling ablity and highlight results ranked **first** and second in that column.

| | MATH | | | AMC 23 | | | AIME 2024 | | | AIME 2025 | | |
|---|---|---|---|---|---|---|---|---|---|---|---|---|
| | 1 | 32 | 64 | 1 | 32 | 64 | 1 | 32 | 64 | 1 | 32 | 64 |
| Base | 14.22 | 50.72 | 57.40 | 4.92 | 50.25 | 62.50 | 0.10 | 3.33 | 6.67 | 0.21 | 5.85 | 10.00 |
| SFT | 30.10 | 72.89 | 76.60 | 15.27 | 58.92 | 65.00 | 0.78 | 8.24 | 10.00 | 1.93 | 19.91 | **23.33** |
| SFT w/ Div | 27.52 | 73.98 | 77.40 | 14.49 | 64.78 | **75.00** | 0.52 | 11.08 | 16.67 | 1.82 | 17.71 | 20.00 |
| SFT w/ Noise | 30.65 | 74.68 | 78.60 | 17.07 | 67.13 | **75.00** | 1.15 | 14.94 | 20.00 | 1.25 | 16.46 | 20.00 |
| REPRISM w/o InN | 27.93 | 74.53 | 78.60 | 15.74 | 69.61 | **75.00** | 0.94 | 13.38 | 16.67 | 1.30 | 17.70 | **23.33** |
| REPRISM | 28.20 | 74.42 | **79.00** | 14.65 | 66.14 | **75.00** | 0.89 | 16.70 | **23.33** | 1.30 | 18.22 | **23.33** |

Table 3: Pass@N performance on code benchmarks.

| | HumanEval | | | HumanEval+ | | | MBPP | | | MBPP+ | | |
|---|---|---|---|---|---|---|---|---|---|---|---|---|
| | 1 | 128 | 256 | 1 | 128 | 256 | 1 | 128 | 256 | 1 | 128 | 256 |
| Base | 45.53 | 84.92 | 87.73 | 39.37 | 76.85 | 79.88 | 46.23 | 77.98 | 80.16 | 46.93 | 72.30 | 74.34 |
| SFT | 50.55 | 91.12 | 93.25 | 46.17 | 85.70 | 87.80 | 54.78 | 88.54 | 89.95 | 47.85 | 80.05 | 82.01 |
| SFT w/ Div | 54.22 | 92.25 | 93.25 | 49.12 | 87.06 | 87.80 | 56.66 | 90.33 | **91.53** | 50.33 | 79.49 | 81.48 |
| SFT w/ Noise | 49.10 | 89.45 | 91.41 | 45.10 | 85.31 | 86.59 | 53.53 | 88.33 | 90.21 | 48.15 | 79.30 | 80.95 |
| REPRISM w/o InN | 53.74 | 93.15 | 93.87 | 48.44 | 87.78 | 89.02 | 54.41 | 88.87 | 90.74 | 49.60 | 78.70 | 80.42 |
| REPRISM | 53.79 | 93.31 | **95.09** | 49.41 | 87.80 | **89.63** | 54.35 | 89.24 | 91.01 | 49.59 | 80.03 | **82.28** |

**Language Model.** We can decompose the per-instance log-probability term in Equation 5 as

$$p_\theta(y(x, \mathbf{N})|\mathbf{E}'_x) = p_\theta(Y = y(x, \mathbf{N})|Y \in \mathcal{D}^x_y, \mathbf{E}'_x) \cdot p_\theta(Y \in \mathcal{D}^x_y|\mathbf{E}'_x). \quad (7)$$

Taking the negative logarithm, we can prove that the per-instance loss term can be decomposed into a component ensuring general answer correctness and another that promotes the noise-conditioned selection of specific diverse answers:

$$-\log p_\theta(y(x, \mathbf{N})|\mathbf{E}'_x) = \underbrace{-\log(\sum\nolimits_{y' \in \mathcal{D}^x_y} p_\theta(y'|\mathbf{E}'_x))}_{\text{Semantic Correctness}} \underbrace{-\log p_\theta(Y = y(x, \mathbf{N})|Y \in \mathcal{D}^x_y, \mathbf{E}'_x)}_{\text{Noise-Conditioned Specificity}} \quad (8)$$

The semantic correctness term ensures that the model consistently generates outputs from the set of correct solutions. The noise-conditioned specificity term pushes the model to use the injected noise as a control signal to select a particular trajectory from the correct set. At test time, by varying the noise injected, we can deterministically and flexibly generate diverse valid outputs for the same input, thus achieving test-time scaling. A detailed explanation of this part is provided in Appendix C.

## 4 EXPERIMENTS

### 4.1 EXPERIMENTS SETUP

**Benchmarks.** Our comprehensive evaluation utilizes 9 challenging benchmarks spanning Math, Code, and Agent tasks. For **code** tasks, we employ the HumanEval, MBPP, HumanEval+, and MBPP+ (Chen et al., 2021; Austin et al., 2021; Liu et al., 2023). HumanEval and MBPP focus on Python code synthesis, and HumanEval+ and MBPP+ are improved versions of them with more rigorous test cases. For **math** tasks, we utilize MATH (Lightman et al., 2023), AMC23, and AIME 24/25 (Mathematical Association of America, 2024), which are problems from high school mathematics competitions. For **agent** tasks, we use ToolQA (Zhuang et al., 2023), which has an easy and a hard split. It covers 8 distinct domains for both splits; we use 6 of them, excluding SciREX and Agenda subsets that rely on an outdated and unstable text retriever.

**Models.** Our experiments are based on the `Llama-3.1-8B-Base`. Among all tasks, we evaluate our method and several baselines: (1) **Base**: `Llama-3.1-8B-Base` with a warm-up SFT to equip it with question answering and instruction following ability. The training data are sampled from the Ultra-Chat dataset (Ding et al., 2023) and math, coding, and agent data distilled from

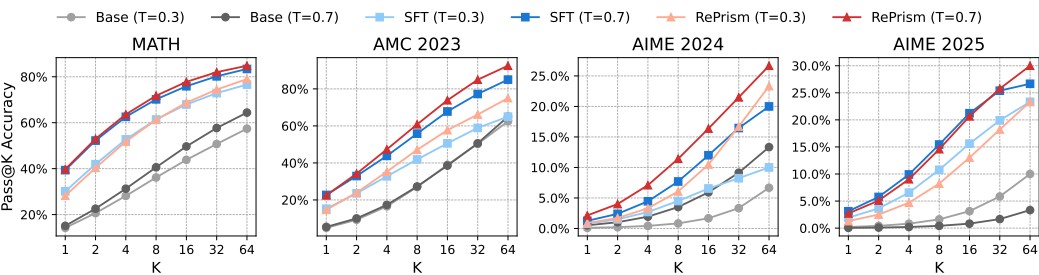

Figure 2: Pass@N performance of REPRISM, SFT, and Base models under different temperatures. REPRISM's performance gain is consistent across different temperatures.

`Llama-3.1-8B-Instruct`. (2) **SFT**: Model fine-tuned using standard supervised methods. (3) **REPRISM**: The complete REPRISM framework, leveraging diverse training data, noise embeddings during training and testing. We also conduct **ablation studies** on code and math tasks under the following settings: (1) **SFT w/ Div**: Fine-tuning performed on datasets with diverse solutions per prompt. (2) **SFT w/ Noise**: Fine-tuning that incorporates noise injection in both training and testing time. (3) **REPRISM w/o InN**: Our full training pipeline (diverse data and training-time noise) but excluding the noise embedding during inference.

**Hyperparameters.** For REPRISM's noise injection, the $\sigma$ was set to $0.001$. Detailed training hyperparameters are in Appendix D. The default temperature for inference is $0.3$.

**Evaluation.** We use the standard unbiased method to calculate pass@N, i.e., $1 - \frac{\binom{N-c}{k}}{\binom{N}{n}}$[1], where $N$ is the total number of samples, $c$ is the number of correct solutions, and $n$ is the number of samples we evaluate (Chen et al., 2021). We calculate Pass@N for each problem individually and then take the average across all problems in a dataset as the overall Pass@N for that dataset.

## 4.2 TRAINING DATA PREPARATION

**Math.** A-m-team[2] provides verified data of different domains distilled from different models (Tian et al., 2025). We utilize the math problems distilled from DeepSeek-R1, R1-distilled models, and Qwen3 series models. For each problem, we collect a maximum of 16 correct answers. We select 10000 problems that have $33\% \sim 67\%$ accuracy among different models, unify their formats, and use the n-gram-based submodular optimization method to sample 4 answers per problem to construct our training dataset. We randomly sample 4 problems for the same questions to form the baseline dataset.

**Code.** For code tasks, our training data was collected from 4 sources: (1) Code data from a-m-team, similar to the math dataset. (2) We gathered around 3100 problems from LeetCode, each featuring multiple human-written solutions. These were processed using DeepSeek-V3 to ensure suitability for training, resulting in an average of 32 solutions per problem; we then selected 8 among them for training. (3) We used problems and test cases from the 'educational instruct' split of the OpenCoder (Huang et al., 2024) SFT-Stage-2 dataset, and generated solutions with Qwen-2.5-Coder-32B. We retained only sampled solutions from problems that have a pass rate lower than $90\%$ and selected 4 solutions per problem. (4) To maintain general coding ability, we integrate problems from 'Evol-Instruct', 'McEval-Instruct', and 'Package-Instruct' splits from OpenCoder SFT-Stage-2. We use both n-gram and AST features for submodular optimization sampling.

**Agent.** We utilized the framework and environment of ToolQA to construct a new set of 2000 training problems. For each problem, we rolled out with different models to generate a maximum of 32 solutions and retain 4 among them. We also use n-gram for sampling here.

---

[1] `https://github.com/openai/human-eval/blob/master/human_eval/evaluation.py`
[2] `https://huggingface.co/a-m-team`

Table 5: Models' performance on math benchmarks before and after RL under different temperatures.

| Base | Phase | Temperature = 0.3 | | | | | | Temperature = 0.7 | | | | | |
|---|---|---|---|---|---|---|---|---|---|---|---|---|---|
| | | AIME24 | | AIME25 | | AMC23 | | AIME24 | | AIME25 | | AMC23 | |
| | | 1 | 64 | 1 | 64 | 1 | 64 | 1 | 64 | 1 | 64 | 1 | 64 |
| SFT | SFT | 0.78 | 10.00 | 1.93 | 23.33 | 15.27 | 65.00 | 1.30 | 33.33 | 3.12 | 26.67 | 22.70 | 85.00 |
| | RL | 5.05 | 33.33 | 4.90 | 23.33 | 34.06 | 92.50 | 5.52 | 33.33 | 5.52 | 33.33 | 36.56 | 87.50 |
| REPRISM w/o InN | SFT | 0.94 | 16.67 | 1.30 | 23.33 | 15.74 | 75.00 | 2.03 | 26.67 | 2.92 | 26.67 | 22.34 | 92.50 |
| | RL | 5.57 | 40.00 | 6.09 | 26.67 | 37.89 | 95.00 | 5.52 | 33.33 | 5.83 | 36.67 | 40.08 | 87.50 |
| REPRISM | SFT | 0.89 | 23.33 | 1.30 | 23.33 | 14.65 | 75.00 | 2.14 | 26.67 | 2.71 | 30.00 | 22.50 | 92.50 |
| | RL | 5.16 | 36.67 | 6.20 | 33.33 | 36.09 | 95.00 | 6.04 | 43.33 | 5.99 | 33.33 | 38.83 | 92.50 |

## 4.3 RESULTS

We present the results of REPRISM compared to base-
line methods and ablations across code, math and
agent tasks in Table 2, Table 3, Table 4, and Figure 2.

In math tasks, our method excels at higher sampling
counts, especially on challenging AMC/AIME prob-
lems. REPRISM more than doubles the pass@64
performance of the standard SFT model on AIME
2024. In code tasks, REPRISM sets new SOTAs, es-

Table 4: Performance on agent tasks.

| Model | Easy | | Hard | |
|---|---|---|---|---|
| | 1 | 32 | 1 | 32 |
| Base | 0.47 | 4.80 | 0.32 | 2.14 |
| SFT | 60.29 | 67.41 | 25.28 | 31.36 |
| REPRISM | 60.55 | 68.01 | 22.55 | 31.74 |

pecially on more rigorous benchmarks like HumanEval+, indicating our models generate not only
more diversed but also more robust solutions. The benefits extend to agent tasks, where REPRISM
achieves the highest pass@32 scores on both ToolQA-Easy and ToolQA-Hard.

**Effectiveness of Diverse Training Data.** Compared to the standard SFT baseline, SFT w/ Div
showed a clear improvement in performance at higher sampling counts. This suggests that exposing
the model to a variety of correct reasoning paths during training successfully broadens its under-
standing of the solution space. However, this approach alone was not sufficient to reach the peak
performance of our complete framework.

**Effectiveness of Noisy Special-token Embeddings.** This mechanism's impact was evaluated
during both training and inference. (1) **Noise as a Training Regularizer**: SFT w/ Noise was
trained on a non-diverse dataset but with our noise injection technique applied. This generally led
to performance improvements over the SFT baseline across various benchmarks. It suggests that
training-time noise can act as a useful regularizer and may encourage the model to learn more robust
representations, as theorized in Section 3.3. (2) **Inference-Time Noise as a Diversity Signal**: The
comparison between RePrism w/o InN and RePrism reveals that introducing noise at inference time
provides a clear boost on diversity, notably on AIME 24 and Humaneval. This strongly indicates
that while the training process teaches the model a diverse set of potential reasoning paths, the
inference-time noise is the key that successfully prompts the model to explore them on demand.

**The influence of Temperature.** Figure 2 compares REPRISM with the standard SFT and base mod-
els under different temperatures on math tasks. REPRISM's performance is consistently higher under
both temperatures, demonstrating that REPRISM can enhance performance whenever exploration is
encouraged by high-temperature sampling. This suggests that REPRISM's exploration-enhancing
mechanism is complementary to, rather than redundant with, temperature scaling.

## 4.4 REINFORCEMENT LEARNING

In RL training, a lack of rollout diversity causes saturation. Since our algorithm generates diversified
thinking patterns, we test REPRISM's effectiveness in the context of RL on math and code tasks.

We use veRL (Sheng et al., 2025) as the training framework. For math reasoning tasks, the training
dataset is sourced from SimpleRL-Zoo (Zeng et al., 2025b). For code generation tasks, we adopt
Acecoder-V2 (Zeng et al., 2025a). Details of RL training are in Appendix E.

Table 6: Models' performance on code benchmarks before and after RL under different temperatures. ↑ and ↓ indicate pass@64 performance improvement and decrease after RL, respectively.

| Base | Phase | Temperature = 0.3 | | | | | | Temperature = 0.7 | | | | | |
| | | HumanEval | | | HumanEval+ | | | HumanEval | | | HumanEval+ | | |
| | | 1 | 8 | 64 | 1 | 8 | 64 | 1 | 8 | 64 | 1 | 8 | 64 |
| --- | --- | --- | --- | --- | --- | --- | --- | --- | --- | --- | --- | --- | --- |
| SFT | SFT | 50.55 | 77.33 | 89.12 | 46.17 | 71.79 | 83.65 | 45.61 | 80.18 | 93.66 | 41.21 | 73.23 | 87.12 |
| | RL | 69.52 | 82.35 | 88.34↓ | 63.63 | 77.49 | 82.32↓ | 68.66 | 85.99 | 92.02↓ | 62.83 | 80.64 | 84.76↓ |
| REPRISM w/o InN | SFT | 53.74 | 80.08 | 91.79 | 48.44 | 74.08 | 85.96 | 47.31 | 81.29 | 93.93 | 42.09 | 75.14 | 88.18 |
| | RL | 69.04 | 83.54 | 88.34↓ | 62.80 | 76.81 | 80.49↓ | 68.11 | 85.61 | 92.64↓ | 62.77 | 79.90 | 86.59↓ |
| REPRISM | SFT | 53.79 | 79.87 | 91.39 | 49.41 | 75.16 | 85.86 | 47.27 | 80.41 | 93.01 | 42.34 | 74.87 | 87.20 |
| | RL | 67.32 | 82.21 | 92.02↑ | 61.80 | 75.57 | 85.37↓ | 67.46 | 86.21 | 93.25↑ | 61.48 | 79.97 | 89.02↑ |

Table 7: Pass@N performance on code benchmarks. REPRISM demonstrates superior sampling efficiency compared with both GEM and DBS.

| | HumanEval | | | HumanEval+ | | | MBPP | | | MBPP+ | | |
| | 1 | 8 | 64 | 1 | 8 | 64 | 1 | 8 | 64 | 1 | 8 | 64 |
| --- | --- | --- | --- | --- | --- | --- | --- | --- | --- | --- | --- | --- |
| Base | 45.53 | 66.94 | 81.14 | 39.37 | 59.19 | 72.89 | 46.23 | 65.29 | 75.42 | 46.93 | 61.74 | 69.99 |
| Instruct | 66.44 | 81.13 | 88.49 | 60.48 | 76.19 | 83.27 | 69.01 | 79.37 | 86.05 | 61.99 | 71.27 | 76.42 |
| DBS on Base | 35.73 | 60.09 | 62.58 | 30.61 | 54.02 | 56.10 | 37.81 | 64.54 | 66.40 | 37.80 | 59.47 | 61.64 |
| DBS on Instruct | 58.29 | 78.37 | 79.75 | 51.56 | 71.08 | 73.17 | 56.26 | 81.92 | 85.45 | 44.41 | 71.14 | 74.87 |
| GEM | 46.03 | 75.90 | 89.57 | 42.20 | 70.12 | 82.32 | 46.21 | 72.51 | 84.13 | 43.01 | 66.37 | 75.66 |
| REPRISM | 49.10 | 78.73 | 87.81 | 45.10 | 74.13 | 83.82 | 53.53 | 76.65 | 86.11 | 48.15 | 68.41 | 77.26 |

Our RL experiments are based on checkpoints obtained from SFT stage. Our experiments settings are as follows: (1) Upon SFT checkpoints, we conduct standard RL training. (2) Upon RePrism checkpoints, we conduct standard RL training without noisy embedding in training or testing stages. (3) Upon RePrism checkpoints, we add noisy embeddings in both training and testing stages.

Table 5 and Table 6 presents the performance of different models before and after RL. We observe the following findings: (1) On math tasks, RePrism w/o InN generally surpasses SFT on small pass after RL, even though its starting point is sometimes lower. This suggests that diversified thinking patterns inherited from the base models can allow the model to benefit more from subsequent RL training. (2) On large-pass metrics, RePrism remains the strongest across various settings. Notably, RePrism is the only variant able to consistently maintain or even enhance its large-pass performance after RL. In contrast, both SFT and RePrism w/o InN typically experience a decrease in pass@64 after RL in code tasks. This highlights the role of noisy embeddings in preserving generation diversity and preventing performance collapse on high-pass metrics during RL training.

Noatbly, the influence of RL exhibit a discrepancy across the two scenarios. We attribute this to the distinct of the tasks, and the different distributional shifts between the training and test data.

## 5 ANALYSES

### 5.1 COMPARING WITH DIVERSITY ENHANCING BASELINES

We also compare our methods with other methods that focus on diversity, on code tasks.

**Baselines.** (1) Diverse Beam Search (DBS) (Vijayakumar et al., 2016): A beam search-based method that optimizes a diversity-augmented objective. We applied DBS to both `Llama-3.1-8B-Base` with warm-up and `Llama-3.1-8B-Instruct`. (2) GEM (Li et al., 2024): GEM is a SFT algorithm that prevent over-memorization and promote diversity in the model's outputs. We applied GEM to the `Llama-3.1-8B-Base` with randomly-sampled training data and default training parameters.

**Results.** As shown in Table 7, our analysis reveals several key findings. (1) DBS consistently degrades performance on both base and instruct models. It results in lower scores across all Pass@1 and Pass@N metrics compared to its respective original model. (2) REPRISM demonstrates superior performance especially at higher sampling budgets. While GEM performs well on HumanEval, our

method achieves the highest Pass@64 scores on the other three benchmarks, outperforming both GEM and the extensively trained `Llama-3.1-8B-Instruct` model.

## 5.2 SOURCING THE EFFECTIVENESS OF REPRISM

To elucidate how REPRISM improves Pass@N performance, we conduct analyses on code tasks. We annotate the algorithms utilized in the solutions generated by REPRISM and SFT models: we firstly categorize all possible solution algorithms into 39 classes and employ the Qwen3-8B model to automatically label each generated solution. The details for labeling is in Appendix G

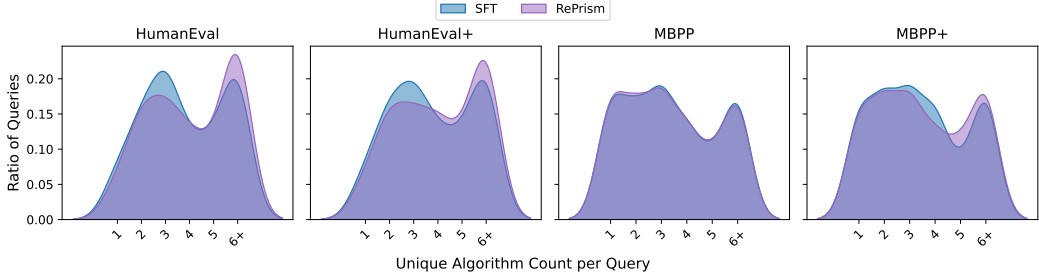

Figure 3: Distribution of unique algorithm counts per query for SFT and REPRISM. The $x$-axis indicates the number of unique algorithm per query, while the $y$-axis shows the proportion of queries corresponding to each count. The distribution for REPRISM consistently shifts toward higher counts, indicating higher algorithmic diversity.

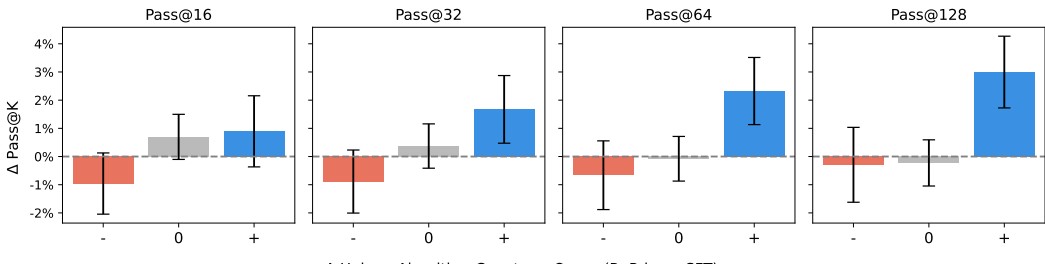

Figure 4: Average Change in Pass@N Grouped by Change in Algorithmic Diversity. Queries are categorized into three groups based on the change in the number of unique algorithms (REPRISM vs. SFT): decrease (-), no change (0), and increase (+). The y-axis represents the change in Pass@N ($\Delta$ Pass@N), displayed as percentages. The height of each bar indicates the mean $\Delta$ Pass@N for queries within its corresponding category. Error bars represent the standard error of the mean.

**REPRISM Enhances Algorithmic Diversity.** For each query, we compute the change in both unique algorithm count and Pass@N score between REPRISM and SFT. As shown in Figure 3, except for MBPP, REPRISM consistently shifts the distribution towards a higher count of unique algorithms per query, which indicates a greater algorithm-level diversity in its outputs compared to SFT.

**Algorithmic Diversity Drives Test-Time Scaling.** We then statistically examine the link between this enhanced diversity and performance gains. For each query, we compute the delta in both the unique algorithm count and the Pass@N score between the REPRISM and SFT models. As illustrated in Figure 4, across all Pass@N settings, an increase in algorithmic diversity (+) correlates with significant improvements in Pass@N, while a decrease (-) is associated with performance drops. This pattern holds robustly across different sample sizes, and the positive impact of increased algorithmic diversity becomes more pronounced as $k$ grows. It indicates that enhanced algorithmic diversity is a primary driver of REPRISM's superior test-time scaling, and enhanced algorithmic diversity is especially beneficial when more samples are available.

### 5.3 ERROR ANALYSIS: THE TRADE-OFF BETWEEN STABILITY AND DIVERSITY

To investigate the underlying causes of REPRISM's lower Pass@1 but higher potential Pass@N under some benchmarks, we conducted a detailed qualitative analysis of the generated solution trajectories. We found that REPRISM's higher error rate in single-pass generation is primarily attributable to its aggressive tendency to explore *heuristic shortcuts* or employ *overly complex tools*, leading to execution failures in valid but difficult reasoning paths. Conversely, the Baseline model tends to adhere to conservative, standard procedures.

Following, we demonstrate a case study of an optimization problem from MATH500, where REPRISM exhibits its largest performance deficit compared to standard SFT regarding pass@1.

---

**Case Study: Minimum Distance to a Parabola**

**Problem Statement:** Find the shortest distance from the origin $(0, 0)$ to the graph of $y = \frac{1}{\sqrt{2}}(x^2 - 3)$.

**Model Performance Comparison:**

- **Baseline (4/10, 40% Accuracy):** Exhibited strong *path dependency*. 10/10 solutions employed the standard Substitution Method $(y(x))$. Errors were strictly computational (e.g., derivative calculation), not methodological.

- **REPRISM (1/10, 10% Accuracy):** Exhibited strong *divergence*. Failed solutions were not due to simple calculation errors, but due to selecting high-risk strategies:
    1. *Heuristic Fallacy:* Assuming the vertex is the closest point.
    2. *Complexity Overload:* Attempting Lagrange Multipliers for a simple quadratic problem.
    3. *Transformation Error:* Variable substitution $t = x^2$ leading to domain constraint errors.

---

As illustrated in the case study above, the Baseline model behaves akin to a rote learner: it applies a single, generic "brute-force" method (substituting $y$ into the distance formula) repeatedly. While calculation errors limited its accuracy to 40%, its strategic consistency was absolute.

In contrast, REPRISM demonstrates a distinct behavioral pattern characterized by **Methodological Diversity** at the cost of stability. Instead of adhering to the standard substitution path, REPRISM attempted four distinct mathematical approaches across ten samples.

This analysis confirms that REPRISM's lower Pass@1 is not a result of random noise, but a side effect of its exploratory nature. By "thinking outside the box", attempting heuristics and advanced tools, it increases the likelihood of stumbling into invalid paths or execution complexities that lead to failure. However, this same mechanism generates valid, novel solution paths that are structurally distinct from the Baseline's clusters. This diversity is the key driver behind REPRISM's superior Pass@N performance, as it covers a broader region of the solution space.

## 6 CONCLUSION AND DISCUSSION

We presented REPRISM, a novel framework designed improves the test-time sampling efficiency of SFT models. By integrating diverse training data sampling via submodular optimization and implicit diversity signals through noisy embeddings, REPRISM enables LLMs to generate a broad spectrum of correct and distinct reasoning paths for a given input. Our extensive experiments across math, code, and agent benchmarks demonstrate that REPRISM consistently delivers superior sampling efficiency. We also demonstrated that REPRISM provides a superior foundation for RL, uniquely enabling models to enhance performance on high-pass metrics where baseline methods often experience degradation.

However, our approach presents a clear trade-off. While REPRISM excels at boosting pass@N performance for larger N, this often comes at the cost of a slight reduction in single-attempt accuracy compared to standard SFT. This suggests that encouraging the model to explore a diverse solution space can sometimes detract from its ability to pinpoint the single most likely correct answer on the first try. Future work could explore adaptive techniques to mitigate this issue, potentially developing mechanisms that can dynamically balance the model's focus between diversity and single-answer precision depending on the task or available computational budget.

## REPRODUCIBILITY STATEMENT

We are committed to comprehensive reproducibility. At the **dataset** level, we use publicly available datasets and official evaluation scripts. For **training and inference**, we provide detailed information on the frameworks, parameters, and other relevant settings in Section 4, Appendix D and Appendix E. For our methodological innovations, we have included the **code** in the supplementary materials.

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

# A SUBMODULAR OPTIMIZATION FOR DIVERSE TRAJECTORY SAMPLING

## A.1 FEATURE ENGINEERING FROM REASONING TRAJECTORIES

The efficacy of our diversity framework hinges on a representation of each candidate's reasoning trajectory $s_k \in S$ through a set of discrete "reasoning features," denoted $F(s_k)$. The nature of these features is domain-specific. For natural language, such as textual explanations or chains of thought, $F(s_k)$ comprises the set of unique n-grams constituting $s_k$. These n-grams serve as proxies for local lexical, syntactic, and semantic patterns, allowing the diversity metric to quantify the variety of expressed concepts and phrasings. In the code generation, where trajectories might represent sequences of operations or derived code structures, features $F(s_k)$ are extracted from Abstract Syntax Trees (ASTs). Such features capture the structural and logical idioms of the code, enabling the selection of a functionally diverse set of trajectories. In all cases, $F(s_k)$ is a finite set of discrete elements for any given $s_k$.

## A.2 THE SUBMODULAR OPTIMIZATION PROBLEM

The core of our approach is the maximization of a set function $f : 2^S \to \mathbb{R}_{\geq 0}$, which quantifies the diversity of a selected subset $A \subseteq S$. This objective is defined as the total number of unique features covered by the trajectories in $A$:

$$f(A) = \left| \bigcup_{s_a \in A} F(s_a) \right|.$$

We now formally establish two key properties of this function: monotonicity and submodularity.

**Proposition 1.** *The function $f$ is monotone non-decreasing.*

*Proof.* Consider any two sets $A, B$ such that $A \subseteq B \subseteq S$. By the properties of the union operator, it follows directly that $\bigcup_{s_a \in A} F(s_a) \subseteq \bigcup_{s_b \in B} F(s_b)$. Taking the cardinality of both sides of this set inclusion, we obtain $\left| \bigcup_{s_a \in A} F(s_a) \right| \leq \left| \bigcup_{s_b \in B} F(s_b) \right|$, which by definition is $f(A) \leq f(B)$. $\square$

**Proposition 2.** *The function $f$ is submodular.*

*Proof.* A set function $f$ is submodular if for any $X \subseteq Y \subseteq S$ and any element $s \in S \setminus Y$, it satisfies the condition of diminishing returns: $f(X \cup \{s\}) - f(X) \geq f(Y \cup \{s\}) - f(Y)$.

Let us examine the marginal gain of adding an element $s$ to an arbitrary set $A \subseteq S$. The gain is given by:

$$f(A \cup \{s\}) - f(A) = \left| F(s) \cup \left( \bigcup_{s_a \in A} F(s_a) \right) \right| - \left| \bigcup_{s_a \in A} F(s_a) \right| = \left| F(s) \setminus \left( \bigcup_{s_a \in A} F(s_a) \right) \right|.$$

This is precisely the number of new features in $s$ that are not already covered by the set $A$.

Now, let $U_X = \bigcup_{s_x \in X} F(s_x)$ and $U_Y = \bigcup_{s_y \in Y} F(s_y)$ be the feature sets covered by $X$ and $Y$, respectively. The submodularity condition is equivalent to proving that $|F(s) \setminus U_X| \geq |F(s) \setminus U_Y|$.

Since $X \subseteq Y$, we have $U_X \subseteq U_Y$. This implies that any feature not covered by $U_Y$ is also not covered by $U_X$. Formally, this gives the set inclusion $F(s) \setminus U_Y \subseteq F(s) \setminus U_X$. Consequently, their cardinalities must satisfy $|F(s) \setminus U_Y| \leq |F(s) \setminus U_X|$. This concludes the proof. $\square$

## A.3 THE GREEDY SELECTION ALGORITHM

To select a subset $A_n$ of size $n$ that maximizes the defined submodular function $f(A)$, we employ a standard greedy algorithm. This iterative procedure builds the solution by repeatedly adding the trajectory that offers the greatest marginal increase in $f(A)$ relative to the set of trajectories selected thus far. Algorithm 1 provides a formal pseudocode description of this process.

---

**Algorithm 1** Greedy Algorithm for Submodular Feature Coverage

---

1: **Input:** Candidate set of trajectories $S = \{s_1, \ldots, s_m\}$; desired subset size $n$.
2: **Input:** Feature sets $F(s_k)$ for each $s_k \in S$.
3: **Output:** A diverse subset $A_n \subseteq S$ of size $n$.
4: $A \leftarrow \emptyset$                                              ▷ Initialize selected set
5: $U \leftarrow \emptyset$                                    ▷ Initialize set of covered features
6: **for** $t \leftarrow 1$ to $n$ **do**
7:     $s^* \leftarrow$ null
8:     max_marginal_gain $\leftarrow -1$
9:     **for** each $s_i \in S \setminus A$ **do**
10:         current_gain $\leftarrow |F(s_i) \setminus U|$
11:         **if** current_gain > max_marginal_gain **then**
12:             max_marginal_gain $\leftarrow$ current_gain
13:             $s^* \leftarrow s_i$
14:         **end if**
15:     **end for**
16:     **if** $s^* =$ null and $|A| < n$ **then**
17:         ▷ No trajectory adds new features, or $S \setminus A$ is empty. Pick any remaining from $S \setminus A$ if needed to reach size $n$.
18:         $s^* \leftarrow$ any element from $S \setminus A$
19:         **if** $s^* =$ null **then**
20:             **break**                      ▷ Cannot select more distinct items
21:         **end if**
22:     **end if**
23:     $A \leftarrow A \cup \{s^*\}$
24:     $U \leftarrow U \cup F(s^*)$
25: **end for**
26: **return** $A$

---

The algorithm iteratively identifies the trajectory $s^*$ that contributes the largest number of features not yet present in the union $U$ of features covered by trajectories already in $A$. The first trajectory selected, $s_1$, is the one possessing the maximum number of unique features, $|F(s_1)|$. Each subsequent trajectory is chosen based on its maximal contribution of new features to the evolving set $U$.

### A.4 THEORETICAL PERFORMANCE GUARANTEE AND ITS SIGNIFICANCE

The utilization of a greedy strategy for maximizing a non-negative, monotone, and submodular set function $f(A)$ subject to a uniform cardinality constraint $|A| \leq n$ is well-justified by strong theoretical results in combinatorial optimization. Nemhauser et al. (1978) provides a constant-factor approximation guarantee for this greedy algorithm. Specifically, the solution $A_n$ obtained by the greedy algorithm satisfies $f(A_n) \geq (1 - 1/e)f(A_{OPT})$, where $A_{OPT}$ represents the true optimal subset of size $n$ that maximizes $f(A)$.

## B PROOF OF THE REGULARIZATION EFFECT OF NOISE INJECTION

Let $\mathbf{E}$ be the matrix of embedding parameters. We analyze the effect of adding independent Gaussian noise $\mathbf{n}_i \sim \mathcal{N}(\mathbf{0}, \sigma^2 \mathbf{I})$ to a subset of embedding vectors $\{\mathbf{e}_i\}_{i \in \mathcal{S}}$, where each $\mathbf{e}_i$ is a row in $\mathbf{E}$. The perturbed embedding matrix is denoted by $\mathbf{E}'$, where each corresponding row is $\mathbf{e}_i' = \mathbf{e}_i + \mathbf{n}_i$. Our objective is to compute the expected loss, $\mathbb{E}_{\mathbf{N}}[L(\mathbf{E}')]$, where $\mathbf{N} = \{\mathbf{n}_i\}_{i \in \mathcal{S}}$.

We perform a second-order Taylor expansion of the loss function $L(\mathbf{E}')$ around the noiseless matrix $\mathbf{E}$. Let $\mathbf{g}_i = \nabla_{\mathbf{e}_i} L(\mathbf{E})$ and $\mathbf{H}_{ij} = \nabla_{\mathbf{e}_i} \nabla_{\mathbf{e}_j} L(\mathbf{E})$ denote the gradient and Hessian blocks, respectively, evaluated at $\mathbf{E}$. For notational convenience, we let $\mathbf{H}_i \equiv \mathbf{H}_{ii}$. The expansion is given by:

$$L(\mathbf{E}') = L(\mathbf{E}) + \sum_{i \in \mathcal{S}} \mathbf{g}_i^T \mathbf{n}_i + \frac{1}{2} \sum_{i,j \in \mathcal{S}} \mathbf{n}_i^T \mathbf{H}_{ij} \mathbf{n}_j + O(\|\mathbf{N}\|^3). \tag{9}$$

We now take the expectation of each term with respect to the noise distribution.

Since $L(\mathbf{E})$ is deterministic with respect to the noise, $\mathbb{E}[L(\mathbf{E})] = L(\mathbf{E})$.

The linear term vanishes in expectation, as the noise has zero mean:

$$\mathbb{E}\left[\sum_{i\in\mathcal{S}}\mathbf{g}_i^T\mathbf{n}_i\right] = \sum_{i\in\mathcal{S}}\mathbf{g}_i^T\mathbb{E}[\mathbf{n}_i] = \mathbf{0}. \tag{10}$$

For the quadratic term, we consider the diagonal ($i = j$) and off-diagonal ($i \neq j$) components separately. For the off-diagonal terms, the noise vectors $\mathbf{n}_i$ and $\mathbf{n}_j$ are independent, hence:

$$\mathbb{E}\left[\mathbf{n}_i^T\mathbf{H}_{ij}\mathbf{n}_j\right] = \mathbb{E}[\mathbf{n}_i^T]\mathbf{H}_{ij}\mathbb{E}[\mathbf{n}_j] = \mathbf{0}^T\mathbf{H}_{ij}\mathbf{0} = 0. \tag{11}$$

For the diagonal terms, we employ the cyclic property of the trace. Recall that the covariance matrix of the noise is $\mathbb{E}[\mathbf{n}_i\mathbf{n}_i^T] = \sigma^2\mathbf{I}$.

$$\mathbb{E}\left[\mathbf{n}_i^T\mathbf{H}_i\mathbf{n}_i\right] = \mathbb{E}\left[\text{tr}(\mathbf{H}_i\mathbf{n}_i\mathbf{n}_i^T)\right] = \text{tr}\left(\mathbf{H}_i\mathbb{E}[\mathbf{n}_i\mathbf{n}_i^T]\right) = \text{tr}(\mathbf{H}_i(\sigma^2\mathbf{I})) = \sigma^2\text{tr}(\mathbf{H}_i). \tag{12}$$

Finally, the expectation of the higher-order terms, $\mathbb{E}[O(\|\mathbf{N}\|^3)]$, is of order $O(\sigma^3)$, assuming sufficient smoothness of $L$.

Combining these results, we arrive at the expected loss:

$$\mathbb{E}_{\mathbf{N}}[L(\mathbf{E}')] = L(\mathbf{E}) + \frac{\sigma^2}{2}\sum_{i\in\mathcal{S}}\text{tr}(\mathbf{H}_i) + O(\sigma^3). \tag{13}$$

## C  DETAILED ANALYSIS OF THE LOSS OBJECTIVE DECOMPOSITION

The training objective is the minimization of the Negative Log-Likelihood (NLL) loss over a distribution of inputs $x$, associated noise vectors $\mathbf{z}$, and corresponding target outputs $y_{x,\mathbf{z}}$. The model is conditioned on a noisy input representation $\mathbf{e}' = \mathbf{e}(x) + \mathbf{z}$, where $\mathbf{e}(x)$ is the noiseless embedding of $x$. The loss is formally expressed as:

$$\mathcal{L}(\theta) = \mathbb{E}_{(x,\mathbf{z})}\left[-\log P_\theta(y_{x,\mathbf{z}}|\mathbf{e}(x) + \mathbf{z})\right] \tag{14}$$

To understand the behavior induced by this objective, we decompose the per-instance log-probability. Let $\mathcal{C}(x)$ denote the set of all correct outputs for a given input $x$. By construction, any target $y_{x,\mathbf{z}}$ is an element of $\mathcal{C}(x)$. We can therefore apply the chain rule of probability as follows:

$$P_\theta(Y = y_{x,\mathbf{z}}|\mathbf{e}') = P_\theta(Y = y_{x,\mathbf{z}}|Y \in \mathcal{C}(x), \mathbf{e}') \cdot P_\theta(Y \in \mathcal{C}(x)|\mathbf{e}') \tag{15}$$

Taking the negative logarithm, the per-instance loss $L_{\text{inst.}} = -\log P_\theta(y_{x,\mathbf{z}}|\mathbf{e}')$ separates into two additive components:

$$L_{\text{inst.}}(x,\mathbf{z};\theta) = \underbrace{-\log P_\theta(Y \in \mathcal{C}(x)|\mathbf{e}')}_{L_C(x,\mathbf{z};\theta)} + \underbrace{-\log P_\theta(Y = y_{x,\mathbf{z}}|Y \in \mathcal{C}(x), \mathbf{e}')}_{L_D(x,\mathbf{z};\theta)} \tag{16}$$

Minimizing the total loss $\mathcal{L}(\theta)$ is thus equivalent to jointly minimizing the expectation of these two terms.

**The Correctness Component** ($L_C$). The first term, $L_C$, depends on the total probability mass the model assigns to the set of all correct answers:

$$L_C(x,\mathbf{z};\theta) = -\log\left(\sum_{y'\in\mathcal{C}(x)} P_\theta(y'|\mathbf{e}')\right) \tag{17}$$

To minimize $\mathbb{E}[L_C]$, the model is incentivized to maximize the collective probability of $\mathcal{C}(x)$. This directly encourages the model to learn the underlying semantic constraints of the task, ensuring its generations are valid solutions.

**The Noise-Conditioned Selection Component** ($L_D$). The second term, $L_D$, is the conditional log-likelihood of selecting the specific target $y_{x,\mathbf{z}}$ from within the set of correct answers:

$$L_D(x, \mathbf{z}; \theta) = -\log P_\theta(Y = y_{x,\mathbf{z}} | Y \in \mathcal{C}(x), \mathbf{e}') \tag{18}$$

To minimize $\mathbb{E}[L_D]$, the model must learn a mapping from the noise vector $\mathbf{z}$ to a specific choice of solution within $\mathcal{C}(x)$. For a fixed $x$, consider two distinct noise vectors, $\mathbf{z}_i$ and $\mathbf{z}_j$, which are paired with potentially different targets, $y_{x,\mathbf{z}_i}$ and $y_{x,\mathbf{z}_j}$. If $y_{x,\mathbf{z}_i} \neq y_{x,\mathbf{z}_j}$, the model's conditional distribution $P_\theta(\cdot | Y \in \mathcal{C}(x), \mathbf{e}')$ must be sensitive to the noise to assign high probability to two different outcomes. An output distribution that is invariant to the noise cannot simultaneously optimize for distinct targets. Therefore, this component compels the model to use the noise $\mathbf{z}$ as a signal to generate diverse and varied outputs from the space of correct solutions.

## D  FINTUNE BACKBONE DETAILS

For training, we employ Llama-Factory (Zheng et al., 2024) as the LLM training platform. Table 8 shows our training hyperparameters in supervised fine-tuning.

Table 8: Hyperparameters for supervised fine-tuning.

| Parameter | Value |
| --- | --- |
| Train batch size | 64 |
| Learning rate | 5.0e-5 |
| Number of epochs | 1.0 |
| LR scheduler | cosine |
| Warmup ratio | 0.1 |
| Precision | bf16 |

## E  REINFORCEMENT LEARNING TRAINING DETAILS

**Framework.** For RL training, we employ verl (Sheng et al., 2025) as the training platform. Table 9 and Table 10 show our training hyperparameters in RL stage for math and code tasks, respectively. All trainings are performed on the 8 A800 GPUs.

**Data.** For math training datasets, simpleRL-zoo (Zeng et al., 2025b) categorized their training data into three difficulty levels(Easy, Medium and Hard) with each category containing roughly 8,000 problems. We adopt the Medium category for training. For code training datasets, we adopt AceCode-V2-122K (Zeng et al., 2025a) as the training dataset.

Table 9: Hyperparameters for reinforcement learning on math reasoning.

| Parameter | Value |
| --- | --- |
| Train batch size | 64 |
| mini batch size | 16 |
| n resp per prompt | 16 |
| max prompt length | 800 |
| max response length | 10000 |
| actor learning rate | 1.0e-6 |
| advantage estimator | GRPO |
| use kl in reward | False |
| use kl loss | True |
| kl loss coefficient | 0.001 |
| repetition penalty | 1.05 |
| temperature | 0.7 |
| use dynamic bsz | True |

Table 10: Hyperparameters for reinforcement learning on code generation.

| Parameter | Value |
| --- | --- |
| Train batch size | 256 |
| mini batch size | 64 |
| n resp per prompt | 16 |
| max prompt length | 2000 |
| max response length | 8000 |
| actor learning rate | 1.0e-6 |
| advantage estimator | GRPO |
| use kl in reward | False |
| use kl loss | True |
| kl loss coefficient | 0.001 |
| repetition penalty | 1.00 |
| temperature | 0.7 |
| use dynamic bsz | False |

## F  HUMAN CHECK RESULTS

In this section, we provide detailed information about the protocol for our human evaluation on the diversity of the training datasets.

**Human Evaluators.**   The human evaluators were selected based on their strong academic backgrounds and expertise in mathematical and informatics olympiads. This rigorous background ensured they possessed the necessary skills to critically analyze the logical and methodical structure of each solution, thereby guaranteeing the correctness and reliability of our human evaluation.

**Human Verification Protocols.**   Two **code** solutions are considered distinct methods only if they meet one of the following criteria:

1. **Algorithm.** The core algorithms used in the solutions are different. For example, one solution might use a recursive approach while another uses an iterative one. Similarly, a Breadth-First Search (BFS) is distinct from a Depth-First Search (DFS).

2. **Data Structure.** The solutions employ the same algorithm but apply it to different data structures. For instance, an algorithm designed for a linked list is considered different from one for a graph, as are those tailored for different tree structures.

3. **Complexity.** The solutions have different time complexity. A solution with $O(n)$ complexity is treated as a distinct method from one with $O(\log n)$ complexity.

Two **math** solutions are considered distinct methods only if one of the following criteria is met:

1. **Distinct Theorems or Lemmas:** The final solutions rely on different theorems or lemmas. The use of a theorem in a failed attempt or a side note does not count.

2. **Fundamentally Different Approaches:** The solutions employ clearly different proof techniques or methods, such as mathematical induction versus proof by contradiction.

## G  LLM-AS-JUDGE PIPELINE

**Predefined Algorithms Pool.**   To create a curated and representative set of algorithms for our analysis, we first randomly sampled 1,000 model-generated answers. We then employed Gemini 2.5 Pro as a judge to analyze the algorithmic approach of each answer. This process involved prompting the large language model to identify and categorize the algorithms used. Following this, Gemini was utilized to aggregate the identified algorithms, which ultimately yielded a pool of 39 distinct and commonly used algorithms for our study.

**Algorithms Selection.**   To classify the algorithms used in each answer, we prompted Qwen3-8B to act as a judge, selecting from the algorithms in our established pool. The specific prompt is listed below.

```
You are an expert computer science algorithm analyst. Your task is to analyze a given
code solution for a coding problem and identify the core algorithm used, based on a
predefined list of algorithms.

# CONTEXT

## 1. Coding Problem
{problem}

## 2. Model's Answer
{answer}

## 3. Predefined Algorithm List

1. Brute Force
2. Greedy Algorithms
```

```
3. Dynamic Programming (DP)
4. Kadane's Algorithm
5. Recursion & Backtracking
6. Divide and Conquer
7. Simulation
8. Constructive Algorithms
9. Linear Search
10. Binary Search
11. Sorting Algorithms
12. Graph Traversal
13. Breadth-First Search (BFS)
14. Depth-First Search (DFS)
15. Shortest Path Algorithms
16. Dijkstra's Algorithm
17. Cycle Detection
18. Hashing / Frequency Counting
19. Stack
20. Heap (Priority Queue)
21. Fenwick Tree (Binary Indexed Tree)
22. Disjoint Set Union (DSU / Union-Find)
23. Monotonic Queue
24. Two Pointers / Sliding Window
25. Prefix / Suffix Sum
26. Array / List Manipulation
27. String Manipulation
28. Regular Expressions (Regex)
29. Palindrome Checking
30. Bit Manipulation (Bitmasking)
31. Number Theory
32. Primality Testing
33. Sieve of Eratosthenes
34. Prime Factorization
35. GCD (Greatest Common Divisor)
36. Modular Arithmetic
37. Binary Exponentiation (Exponentiation by Squaring)
38. Number Base Conversion
39. Combinatorics

# TASK

Carefully analyze the provided "Model's Answer" in the context of the "Coding Problem".
 Your goal is to select the algorithm from the "Predefined Algorithm List" that best
describes the solution's methodology.

- If the solution directly implements an algorithm from the list, state its number
before the name.
- If the solution does not use an algorithm with an exact name from the list, identify
the **most relevant or conceptually closest** algorithm. For example, if the solution
uses memoization and the list contains "Dynamic Programming", you should choose "
Dynamic Programming".
- If the solution combines multiple techniques from the list, you should list all
relevant algorithms by their numbers, separated by commas.

# REQUIRED OUTPUT FORMAT

Provide your answer in the following json format, and do not add any extra commentary
before or after.

{{
    "reasoning": "<Brief explanation of why this algorithm was chosen based on the
    solution's approach.",
    "algorithm": "<Selected Algorithm Number from the Predefined List>"
}}
```

```
# EXAMPLE OUTPUT
{{
    "reasoning": "The solution uses a DP table to store intermediate results, which is
    characteristic of 3.Dynamic Programming. Also, it employs recursion to explore
    different solution paths, indicating the use of 5.Recursion & Backtracking.",
    "algorithm": "3,5"
}}
```

## H   LLM USAGE

LLMs were used for grammar checking. However, the core scientific contributions, including the conceptualization of the research, experimental design, analysis, and the final conclusions, are entirely the work of the human authors.

