# OpenReview forum: "Diversity-aware Training for Test-time Scaling"
_ICLR.cc/2026/Conference — Submitted to ICLR 2026_

### Official Review · Reviewer_Ms5P · 2025-10-19

**Soundness:** 2
**Presentation:** 3
**Contribution:** 2
**Rating:** 2
**Confidence:** 3

**Summary:**

This paper introduces REPRISM, a training framework designed to improve the test-time sampling efficiency of LLMs by addressing the issue of diminishing returns caused by a lack of diversity in generated solutions. The method trains models to produce a diverse set of correct reasoning paths using two core components: (1) a specialized training dataset, curated with submodular optimization, that contains multiple diverse and correct solutions for each prompt, and (2) the injection of Gaussian noise into special token embeddings to act as an implicit control signal for diversity. The authors validate REPRISM across 9 math, code, and agent benchmarks, demonstrating minor improvements in pass@k accuracy.

**Strengths:**

1. Extensive empirical studies are done for an in-depth understanding of the proposed method.
2. Figures/tables are clear and well designed

**Weaknesses:**

This paper has numerous weaknesses. Notably, (1) the empirical results are concerning, and (2) the main contribution of this paper is unclear as having more diverse data (quality), and more data (quantity) is already well known in the literature. Noise injection has also been studied extensively.

1. Empirical gains stated in the abstract are minimal, and could very well be from noise

> We validate REPRISM on 9 challenging benchmarks across Math, Code, and Agent tasks, where it increases the models’ pass@N accuracy by up to 6.4%, 1.1%, and 0.5% on Math, Code, and Agent tasks, respectively

2. Significant concern on empirical results. When we deploy a model in the real-world pass@K only works on queries (x) with verifiable answer (y). Therefore, real-world performance is most closely proxied by pass@1. Consider pass@1 results in fig. 1 where the proposed method commonly performs worse than regular SFT. To show that it can be deployed in the real-world for any potential query from a user, it would have been  better to show self-consistency (majority voting) results over pass@k

3. Experiments might be incomplete. Why is Fig 1’s first curve missing data points at x = 128, 256. Why does the last curve (Agent) have no post-RL results?

4. Empirical gains could be from an increased train data set size in terms of total tokens. Authors should provide clear comparison experiments.

5. Are the authors sure that there are no additional baselines? Their only baseline is SFT

6. Figure descriptions are commonly unclear. e.g.

Fig. 1 it is more clear to have said comparison of SFT and RePrism as displayed in the order in the image

> Figure 1: (Left) Comparison of REPRISM and SFT

and some figures/tables would help to have more description.

7. Figure 1 is missing information: no x-axis label is provided.

8. Typos should be fixed. E.g. Fig. 1 agerage → average

9. Writing needs revision. Why does the paper sometime use pass@k and pass@N, arent these referring to the same thing? Let me know if i am mistaken.

10. Writing is overselling the objective empirical results. E.g.

> Beyond improving test-time sampling, we find that the diversity fostered by REPRISM serves as a powerful catalyst for RL.

>  exceptional foundation for RL

The empirical gains are too minor to call it a “powerful catalyst” and “exceptional foundation”

11. Human evaluations do not have IRB approval or other information related to how it was set-up. This can raise ethical concerns like lack of appropriate compensation.

12. Hyperparameters (line 241) seem arbitrary. Why pick these? Are results sensitive to hyperparameters? Because this can be a huge headache for practitioners to tune this hyperparameter.

13. The experiments only use a single LLM (Llama-3.1-8B) making generalization to other model families unclear. This is concerning as there are other highly popular open-src model families like Qwen series

**Questions:**

1. There are a lot of noise injection related literature that improves performance like NEFTune: Noisy Embeddings Improve Instruction Finetuning. Are these not relevant baselines?

2. What is the cost overhead of this method, and how does its overheads compare to existing methods? This seems to be a critical discussion point that i can not find (or have missed)

could be from data curation, or compute, or other economic costs. This should be discussed to get a clear view of the contribution.

**Details Of Ethics Concerns:**

Human subjects are involved.

Requires a check if the human experiments are sound.

Unclear if they were fairly compensated.

---

> ### Author Response · Authors · 2025-11-27
> **Reply to Reviewer Ms5P - 1/2**
>
> Thank you for your thoughtful review. We address your concerns below.
>
> ---
>
>
> ## Magnitude and Practical Meaningfulness of Gains
>
> > Empirical gains stated in the  abstract are minimal, and could very well be from noise
>
> We would like to clarify that on all 8 math and code datasets, our method consistently outperforms SFT in pass@N accuracy, and achieves the best result among all methods (including ablations) in all 8 cases. Notably, the pass@N baselines are already quite high, so further improvements are non-trivial: our method achieves average gains of 6.4% on math and 1.1% on code, which we believe are meaningful in this context. Moreover, all pass@N results are computed using the standard, unbiased estimator widely used in the community, which ensures the gains are not due to randomness or noise. We will include standard deviations in the updated version for further clarity.
>
>  ## Pass@1 vs. Pass@N and Real‑World Deployment
>
> > Significant concern on  empirical results. When we deploy a model in the real-world pass@K only works  on queries (x) with verifiable answer (y). Therefore, real-world performance  is most closely proxied by pass@1. Consider pass@1 results in fig. 1 where the  proposed method commonly performs worse than regular SFT. To show that it can  be deployed in the real-world for any potential query from a user, it would  have been better to show self-consistency (majority voting) results over  pass@k
>
>
> Our method is specifically designed to enhance *exploration* and *diversity* (pass@N), which is crucial in settings where multiple attempts are possible, or when a verifier is available to select correct outputs (such as in code generation, math problem solving, or data curation for self-improving models). In these scenarios, pass@N is the primary metric of interest. While REPRISM sometimes slightly lowers pass@1, this is a well-known trade-off for diversity-promoting techniques. We directly acknowledge this limitation in our conclusion and discuss potential directions for adaptive balancing between accuracy and diversity.
>
>  ## Missing Points in Figure 1 and Agent RL Results
>
> > Experiments might be  incomplete. Why is Fig 1’s first curve missing data points at x = 128, 256.  Why does the last curve (Agent) have no post-RL results?
>
> Due to computational constraints, we used different maximum sample sizes for each task. The agent tasks are substantially slower to evaluate at large sampling budgets, which is why we limited the maximum k for these settings.
>
>  ## Controlling for Total Training Tokens and Method Overheads
>
> > Empirical gains could be from  an increased train data set size in terms of total tokens. Authors should  provide clear comparison experiments.  What is the cost overhead of  this method, and how does its overheads compare to existing methods? This  seems to be a critical discussion point that i can not find (or have missed)
>
> Thank you for raising this point! We want to clarify that all SFT experiments are carefully controlled to use the same amount of training data (total tokens), ensuring that any performance gain is not due to increased data size. Our diversity sampling does not increase the average length of training examples. The main additional cost is a one-time offline submodular selection step (complexity O(K·M·N), see main rebuttal), which is efficient for moderate K and M. Noise injection adds negligible overhead at both training and inference (just simple vector addition on BOS/EOS positions).

---

> ### Author Response · Authors · 2025-11-27
> **Reply to Reviewer Ms5P - 2/2**
>
> ## Baselines Beyond Vanilla SFT
>
> > Are the authors sure that  there are no additional baselines? Their only baseline is SFT  There are a lot of noise  injection related literature that improves performance like NEFTune: Noisy  Embeddings Improve Instruction Finetuning. Are these not relevant baselines?
>
>  We include extensive ablation studies (e.g., SFT w/ Noise, SFT w/ Div) as well as comparisons against state-of-the-art diversity methods such as Diverse Beam Search (DBS) and GEM in Section 5.1. GEM specifically is a recent SOTA for diversity-aware LLM training. Our method outperforms these baselines on most code/math benchmarks. NEFTune-style methods are covered by our "SFT w/ Noise" ablation, which shows that noise alone does not achieve the same gains as our full framework combining data and noise.
>
>  ## Human Evaluation and Ethical/IRB Considerations
>
> > Human evaluations do not have  IRB approval or other information related to how it was set-up. This can  raise ethical concerns like lack of appropriate compensation.
>
>
> All human annotation was performed by the authors themselves; no external annotators or compensation were involved. We have added this clarification to the revised manuscript
>
>  ## Hyperparameter Choices and Sensitivity
>
> > Hyperparameters (line 241)  seem arbitrary. Why pick these? Are results sensitive to hyperparameters?  Because this can be a huge headache for practitioners to tune this  hyperparameter.
>
> σ=0.001 was chosen to be an order of magnitude smaller than the typical embedding scale, based on preliminary experiments. Sensitivity is low within a reasonable range; we will add a hyperparameter sweep in the appendix.
>
>  ## Generalization Beyond LLaMA‑3.1‑8B
>
> > The experiments only use a  single LLM (Llama-3.1-8B) making generalization to other model families  unclear. This is concerning as there are other highly popular open-src model  families like Qwen series
>
> Thank you for the suggestion. We have also run experiments on Qwen2.5-7B. On AIME25@64, SFT achieves 36.7 and Reprism achieves 43.3.
>
> ---
>
> Thank you again for your time and effort. We hope our response resolves your concerns. We welcome any further comments.

---

### Official Review · Reviewer_wGCG · 2025-10-21

**Soundness:** 2
**Presentation:** 2
**Contribution:** 1
**Rating:** 2
**Confidence:** 5

**Summary:**

The paper propose REPRISM, a novel algorithm used on LLM training with SFT and RL to enchance the generation diversity of LLMs by adding noise on embedding space. The algorithm is tested on Llama on math, coding and agent task to show the proposed algorithm can provide a benefit compared to the baselines compared.

**Strengths:**

- The problem of increasing generation diversity in LLM is important.

**Weaknesses:**

- The clarity of the paper can be improved. While the paper include many mathematical notations, a lot of them lack a definition and is unclear what is exactly referring to in the context. Please refer to my questions for some examples.
- The idea is not completely novel, and the related works are greatly missing. Similar idea of manupilating latent space has been explored [1, 2], improving diversity in LLM training without prompt have been explored [3].
- The experiments are on Llama only. More recent models need to be tested.

[1]. Geiping J, McLeish S, Jain N, et al. Scaling up test-time compute with latent reasoning: A recurrent depth approach[J]. arXiv preprint arXiv:2502.05171, 2025.

[2]. Zhang Z, He X, Yan W, et al. Soft thinking: Unlocking the reasoning potential of llms in continuous concept space[J]. arXiv preprint arXiv:2505.15778, 2025.

[3]. Chen W, Zhang Z, Liu G, et al. Flaming-hot Initiation with Regular Execution Sampling for Large Language Models[C]//Findings of the Association for Computational Linguistics: NAACL 2025. 2025: 7118-7127.

**Questions:**

The current paper needs significant improvement in all dimensions mentioned in the weakness section. Especially, additional experiments on more models and compared to different baselines are needed.

Specifically for clarity:
1. What is L? iIs it the number of propmts per batch? Or length of prefix?
2. Is the noise sequence $n$ pre-defined and fixed throughout different prompts?

---

> ### Author Response · Authors · 2025-11-27
>
> Thank you for your thoughtful review. We address your concerns below.
>
> ---
>
> ## Relation to Prior Work
>
> > The idea is not completely  novel, and the related works are greatly missing. Similar idea of  manupilating latent space has been explored [1, 2], improving diversity in  LLM training without prompt have been explored [3].
>
> Thank you for pointing out relevant literature [1,2,3]. While previous works have explored latent space manipulation and diversity, our novelty is in: (a) using noise as an explicit, interpretable diversity control signal, (b) combining it with principled data selection (submodular); and (c) demonstrating synergy between diversity signals and data. We will expand the Related Work section to include these works and clarify distinctions.
>
> ## Clarifying the Definitions
>
> > The clarity of the paper can  be improved. While the paper include many mathematical notations, a lot of  them lack a definition and is unclear what is exactly referring to in the  context. Please refer to my questions for some examples.  What is L? iIs it the number  of propmts per batch? Or length of prefix?
>
> In Section 3.2, L refers to the length of the input sequence (not batch size). The noise sequence is sampled independently for each input at each occurrence (not fixed). Only BOS/EOS tokens are perturbed.
>
> > Is the noise sequence pre-defined and fixed throughout different  prompts?
>
> As described in Section 3.2, we add randomly sampled noise specifically to special tokens such as BOS/EOS for each input. The noise sequence is not pre-defined or fixed across prompts, but is independently sampled each time.
>
> ---
>
> Thank you again for your time and effort. We hope our response resolves your concerns. We welcome any further comments.

---

### Official Review · Reviewer_CKUo · 2025-10-25

**Soundness:** 3
**Presentation:** 3
**Contribution:** 3
**Rating:** 6
**Confidence:** 3

**Summary:**

The paper proposes REPRISM, a training-time framework to improve test-time scaling of LLMs by increasing reasoning-level diversity. It (1) constructs per-prompt training sets with multiple diverse-but-correct solutions via submodular sampling, (2) injects Gaussian noise into selected special-token embeddings as an implicit diversity signal during training and inference, and (3) provides theoretical analysis of stability and the training objective decomposition. Experiments on 9 benchmarks show improved sampling efficiency / pass@N gains and more robust RL fine-tuning behaviour.

**Strengths:**

1. Novel integration of submodular optimization for diverse data sampling and noise embeddings for implicit diversity control;
2. Comprehensive evaluation across 9 benchmarks in math, code, and agent tasks, with consistent improvements in Pass@N;
3. Theoretical analysis showing noise injection acts as regularization and enables controlled diversity;
4. Demonstrates enhanced RL performance and prevents diversity collapse.

**Weaknesses:**

1. Insufficient Reproducibility: Key details are missing, including the random seed, data partitioning, specific rules for selecting special tokens, and explicit settings for the number of sampling trajectories m per prompt (the appendix only lists some hyperparameters).
2. Insufficient Ablation Experiments: Although several variants are reported, they do not adequately demonstrate the contributions of different components across various datasets and sampling budgets k.
3. Insufficient Robustness Analysis: While pass@1 decreases in some cases, there is a lack of detailed analysis regarding the failure scenarios.
4. Strong Dependence on Feature Representations: The approach relies on n-gram/AST features, which may limit sampling quality if these features fail to sufficiently capture the diversity of reasoning.
5. Limited Manual Annotation Scale: Table 1 includes annotations for only 32 queries, indicating a relatively small sample size.

**Questions:**

1. Candidate set construction: for each prompt you mention collecting up to m trajectories. What is the typical m used per dataset / per prompt in experiments, and how sensitive are results to m?
2. Special tokens / noise schedule: which tokens are considered “special tokens” and how were their positions chosen? Is σ=0.001 fixed for all experiments — did you sweep σ and report sensitivity?
3. Reproducibility: authors state code is in supplementary—will the code include the full data-sampling / feature extraction (n-gram, AST) pipeline, LLM prompts, and random seeds? Appendix references but please clarify.
4. Negative impact cases: can you provide example instances where REPRISM reduces pass@1 and analyze why (e.g., overly promoting exploratory but low-probability valid solutions)?

---

> ### Author Response · Authors · 2025-11-27
> **Reply to Reviewer CKUo - 1/2**
>
> Thank you for your thoughtful review. We address your concerns below.
>
> ---
>
> ## Reproducibility: Seeds, Partitioning, and m per Prompt
>
> > Insufficient Reproducibility: Key details are missing, including the random seed, data partitioning, specific rules for selecting special tokens, and explicit settings for the number of sampling trajectories m per prompt (the appendix only lists some hyperparameters).
>
> > Candidate set construction: for each prompt you mention collecting up to m trajectories. What is the typical m used per dataset / per prompt in experiments, and how sensitive are results to m?
>
>
>
> We use the default seed 42 from LLaMA‑Factory for all SFT experiments.
>
> Details of Data partitioning and samples per prompt are already detailed in Section 4.2:
>
> - **Math:** From A‑M‑Team’s verified data, we collect up to **16** correct answers per problem and then use submodular optimization to select **4** solutions per problem.
>
> - **Code:** (1) LeetCode: ~3,100 problems, each with ~32 human solutions. We process them with DeepSeek‑V3 and keep **8** candidates per problem, then submodularly select **4**. (2) Other sources (A‑M‑Team, OpenCoder educational): m≤16 candidates; again we select 4 per prompt.
>
> - **Agent:** For each problem, we roll out up to **32** trajectories with different models and select **4**via submodular optimization.
>
> ## On Special Tokens
>
> > which tokens are considered  “special tokens” and how were their positions chosen?
>
> Special Token is a proper noun in the firld of LLMs. We clarify that **BOS and EOS** are treated as special tokens for noise injection in our experiments.
>
> ## On the Noise Magnitude σ
>
> > Is σ=0.001 fixed for all experiments — did you sweep σ and report sensitivity?
>
> We fix σ = 0.001 for all experiments. This value was chosen so that the noise magnitude is small relative to embedding norms but still visible to the model. We did not perform a wide hyperparameter sweep for σ due to resource constraints.
>
> ## Decomposition of Contributions via Ablations
>
> > Although several variants are reported, they do not adequately demonstrate the contributions of different components across various datasets and sampling budgets k.
>
> We believe our ablations do systematically isolate the contributions of each component and we will clarify this in the text.
>
> Specially, **in SFT experiments**, we have (1) **SFT:** vanilla supervised fine‑tuning, (2) **SFT w/ Div:** diverse‑data only (submodular sampling, no noise), (3) **SFT w/ Noise:** noise only (no diverse sampling), (4) **REPRISM w/o InN:** diverse data + training‑time noise; no inference‑time noise, (5) **REPRISM:** full method.
>
> This design allows us to see, for each dataset and multiple values of k, that (1) Diverse data alone improves pass@N vs SFT, (2) Noise alone improves robustness and sometimes pass@1, (3) The combination yields further improvements, and (4) Inference‑time noise gives an extra boost, especially at large k.
>
>  **In RL experiments**, we start RL from three different SFT checkpoints (SFT, REPRISM w/o InN, REPRISM) and also vary whether to use noise in RL training and testing. This shows that: (2) REPRISM’s more diverse rollouts lead to better RL gains, and (2) Using noise during RL helps avoid collapse of pass@64 in code tasks.
>
> We will make this decomposition more explicit in the ablation sections and add cross‑dataset summaries highlighting how each component behaves as k varies.

---

> ### Author Response · Authors · 2025-11-27
> **Reply to Reviewer CKUo - 2/2**
>
> ## Analysis of Cases Where REPRISM Reduces Pass@1
>
> > While pass@1 decreases in  some cases, there is a lack of detailed analysis regarding the failure  scenarios.  can you provide example  instances where REPRISM reduces pass@1 and analyze why (e.g., overly  promoting exploratory but low-probability valid solutions)?
>
>
>  We thank the reviewer for this suggestion and have conducted a manual case study on prompts where REPRISM underperforms SFT on pass@1 in math and code.
>
> We found a clear pattern: **Baseline SFT** behaves like a very “conservative student”: it tends to follow a single, memorized template solution. This limits exploration, but it also means the model rarely “goes off track”.
>
> In contrast, **REPRISM** often tries to find **shortcuts (heuristics)** instead of fully working through the standard method; these shortcuts can be fragile and produce errors. It sometimes **over‑engineers** the solution (e.g., using unnecessarily complex tools, extra variables, or transformations), which introduces more opportunities for arithmetic or logical mistakes. It also uses **alternative formulations** (e.g., variable substitutions, re‑parameterizations) that add more steps where errors can occur.
>
> So REPRISM’s slightly lower pass@1 stems mainly from its **more exploratory strategy**, not from worse understanding of the problem. This same exploratory tendency, however, is exactly what yields **higher pass@N** and offers a richer search space for RL.
>
> We have added examples and a discussion of these failure modes in the revised paper and explicitly state this exploration–accuracy trade‑off as a limitation.
>
> ## On the N‑gram/AST Features and Modularity
>
> > The approach relies on  n-gram/AST features, which may limit sampling quality if these features fail  to sufficiently capture the diversity of reasoning.
>
> Actually, the specific feature choices (n‑grams, AST fragments) are *not* the core contribution of our work; they are a reasonably effective and simple instantiation. Our framework is modular in the feature representation, where better features for reasoning (e.g., learned representations, proof skeletons, higher‑level program graphs) could be directly plugged into the **same submodular coverage** objective.
>
> We expect that improved features would **further enhance** the diversity and quality of selected trajectories, and thus improve REPRISM’s performance. We will clarify that feature engineering is orthogonal to our main contributions and can be upgraded in future work.
>
>  ## Scale and Purpose of the Human Annotation Study
>
> > Table 1 includes annotations for only 32 queries, indicating a relatively small sample size.
>
>
> Currently, Table 1 is based on manual annotation of 2 domains (math and code) × 2 selection methods (random vs submodular) × 32 queries per domain × 4 solutions per query.
>
> This yields *512 individually annotated solutions. While 32 queries per setting is not large, the total annotation effort is substantial.  Given the costs of careful annotation and the fact that this is only one of several diversity analyses (we also provide automated algorithm‑class labeling on code), we believe the scale is reasonable.
>
> ---
>
> Thank you again for your time and effort. We hope our response resolves your concerns. We welcome any further comments.

---

### Official Review · Reviewer_Sixx · 2025-11-01

**Soundness:** 2
**Presentation:** 2
**Contribution:** 1
**Rating:** 2
**Confidence:** 4

**Summary:**

This work presents two complementary strategies for enhancing inference performance in mathematical reasoning, code generation, and agent-based decision tasks. The first leverages submodular optimization to promote diversity among generated outputs, while the second improves model robustness by injecting noise into the input embeddings during training. Empirical results demonstrate consistent gains in accuracy.

**Strengths:**

1. The method employs a principled submodular approach to select a diverse set of trajectories.
2. Injecting noise into the input embeddings appears to yield practical performance improvements.

**Weaknesses:**

1. There is no significant improvement in pass@N performance for math and code. Standard deviation numbers are not given in those experiments.
2. While n-gram–based diversity metrics are standard for identifying redundant text, they do not necessarily reflect the deeper logical diversity present across trajectories.
3. The inherently sequential nature of submodular optimization may limit the scalability of this approach. This may limit potential gains on more complex tasks.
4. Injecting noise into embeddings also requires applying noise at inference, which may be undesirable. A more suitable approach may be needed.

**Questions:**

1. What is the ceiling performance when you scale up the inference compute / your sampling budget is around infinite?
2. While n-gram diversity may not capture the underlying logical structure of mathematical reasoning, can you provide intuition for why a bag-of-n-grams representation is still a reasonable proxy for diversity among solution trajectories?
3. What is the baseline in Table 1? I.e. randomly sample 4 outputs w/o using submodularity?
4. Can you provide the algorithmic complexity of sumbodular optimization? Assuming you are selecting K trajectories, each with N bag-of-words?
5. It is not entirely clear why perturbing the embeddings during training improves performance. Is the effect primarily due to the regularization it introduces?
6. If perturbing the embeddings during training improves performance, then based on the theory in (6), would applying explicit regularization achieve a similar effect?

---

> ### Author Response · Authors · 2025-11-27
> **Reply to Reviewer Sixx - 1/2**
>
> Thank you for your thoughtful review. We address your concerns below.
>
> ---
>
> ## Pass@N Improvements and Reporting of Variance
>
> > There is no significant improvement in pass@N performance for math and code. Standard deviation numbers are not given in those experiments.
>
> Thank you for your comments. Actually, on all 8 code and math datasets, our method outperforms SFT on pass@N and our method achieves the best pass@N among all methods (including ablations) on 7/8 datasets. Since pass@N baselines are already high, further improvements are challenging; our average gains (math: +6.4%, code: +1.1%) are substantial.
>
> ## On  N‑gram Features as a Proxy for Reasoning Diversity
>
> > While n-gram–based diversity metrics are standard for identifying redundant text, they do not necessarily reflect the deeper logical diversity present across trajectories.
>
> > While n-gram diversity may not capture the underlying logical structure of mathematical reasoning, can you provide intuition for why a bag-of-n-grams representation is still a reasonable proxy for diversity among solution trajectories?
>
> We agree that n‑grams alone are not a perfect representation of deep logical structure. In our framework, they are used within a submodular coverage objective, which explicitly penalizes redundancy and focuses on incremental coverage, not raw n‑gram counts.
>
> In practice, for math and code, n‑grams capture keywords and local structures that correlate with different solution strategies. Under submodular optimization, these “strategy markers” are preferentially selected because they add new features.
> For code, we do not rely on n‑grams alone: we also include AST features, which are much closer to algorithmic structure and help distinguish logically different solutions.
>
> To empirically validate that these features correlate with reasoning‑level diversity, we conducted human annotations on both: (1) Randomly selected sets of 4 solutions per prompt (baseline), and (2)Submodularly selected sets of 4 solutions per prompt (our method).
>
> We counted, for each prompt, how many different solution strategies appear among the 4 solutions (≥3, =2, or =1). The results are:
>
> - Random selection (baseline):
>
> | Dataset | ≥3   | 2    | 1    |
> |--------|------|------|------|
> | Math   | 28.1 | 37.5 | 34.4 |
> | Code   | 25.0 | 37.5 | 37.5 |
>
> - Our submodular selection (reported in the paper):
>
> | Dataset | ≥3   | 2    | 1    |
> |--------|------|------|------|
> | Math   | 42.8 | 39.3 | 17.9 |
> | Code   | 34.4 | 37.5 | 28.1 |
>
> Submodular sampling clearly reduces the “only 1 strategy” cases and increases the “≥3 strategies” bucket, especially on math. This suggests that, combined with the coverage objective and ASTs, our feature representation is indeed a useful proxy for reasoning‑level diversity.
>
> > What is the baseline in Table 1? I.e. randomly sample 4 outputs w/o using submodularity?
>
> Originally Table 1 only reported results with submodular selection. We have now explicitly added the random‑sampling baseline with the diversity statistics.
>
> # The Sequential Nature of Submodular Optimization
>
> > The inherently sequential nature of submodular optimization may limit the scalability of this approach. This may limit potential gains on more complex tasks.
>
> Being sequential is not a drawback. Submodular is a well-studied approach with strong theoretical guarantees: the (1-1/e)-approximation.  Given that the algorithm is only used once during data curation, and not at inference time, we found it scalable enough even for our largest datasets. We have added this complexity analysis to the paper.

---

> ### Author Response · Authors · 2025-11-27
> **Reply to Reviewer Sixx - 2/2**
>
> ## Role of Noise at Training vs. Inference Time
>
> > Injecting noise into embeddings also requires applying noise at inference, which may be undesirable. A more suitable approach may be needed.
>
>  > It is not entirely clear why perturbing the embeddings during training improves performance. Is the effect primarily due to the regularization it introduces?
>
> We understand the concern. In our design, noise has a dual role:
>
> 1. **Training‑time signal**: During training, noise at special tokens acts as an **implicit diversity control signal**. For the same input x, different noise patterns are paired with different correct solutions (submodularly sampled). The model learns to associate **noise variations** with **different valid reasoning paths**.
>
> 2. **Inference‑time knob**: At inference, adding noise at BOS/EOS allows us to **actively steer** the model towards different reasoning modes for the *same* input — which is exactly what we want in test‑time scaling scenarios.
>
> Our ablation **REPRISM w/o InN** uses diverse data and training‑time noise, but **no noise at inference**. It still outperforms standard SFT on pass@N in many settings. Full REPRISM (with inference noise) further improves diversity and pass@N, especially on the most difficult benchmarks (e.g., AIME 2024, HumanEval).
>
> ## Complexity of Submodular Optimization
>
> > Can you provide the  algorithmic complexity of sumbodular optimization? Assuming you are selecting  K trajectories, each with N bag-of-words?
>
>  Given M candidate trajectories per prompt, K to select, and N features per trajectory, the complexity is O(K·M·N). For each of K iterations, we compute the marginal gain for each of up to M candidates, each requiring comparison over N features. In our experiments, M and K are modest, making this efficient in practice.
>
> ## Noise vs. Explicit Regularization
>
> > If perturbing the embeddings during training improves performance, then based on the theory in (6), would applying explicit regularization achieve a similar effect?
>
> Our noise is intended to be an implicit diversity control signal. During training, the model learns to associate different noise seeds (on BOS/EOS) with different valid solutions. During inference, sampling with different noise enables exploration of distinct reasoning paths. Our ablation (no noise at inference) shows that inference-time noise is necessary for full diversity, matching our design intent. Meanwhile, we also include an SFT w/ Noise baseline that treats noise purely as explicit regularization, and we observe that this does not lead to comparable gains in diversity.
>
> ---
>
> Thank you again for your time and effort. We hope our response resolves your concerns. We welcome any further comments.

---

### Meta-Review · Area_Chair_pdUD · 2026-01-08

**Summary:**

The paper proposes REPRISM, a training-time framework to improve test-time scaling of LLMs by increasing reasoning-level diversity. The reviewers raised the following concerns:

1. The technical novelty of the contribution is unclear.

2. The method does not demonstrate significant improvements

Although the proposed approach is promising, the paper is not recommended for acceptance in its current form. The authors are encouraged to address the reviewers’ feedback and further strengthen the work for resubmission to other venues.

**Reviewer Concerns:**

The ideas are not innovative enough, and the experiments are weak.

**Reviewer Scores:**

NA

---

### Decision · Program_Chairs · 2026-01-26

Reject